# Determining the Effect of Pancreatic-like Enzymes (PLEMs) Added to the Feed of Pregnant Sows on Fetal Size of Piglets to Minimize IUGR Syndrome Caused by Fetal Malnutrition

**DOI:** 10.3390/ani13223448

**Published:** 2023-11-08

**Authors:** Marek Pieszka, Paulina Szczurek, Sylwia Orczewska-Dudek, Marian Kamyczek, Magdalena Pieszka

**Affiliations:** 1Department of Animal Nutrition and Feed Science, National Research Institute of Animal Production, 32-083 Kraków, Polandsylwia.orczewska@iz.edu.pl (S.O.-D.); 2National Research Institute of Animal Production, The Institute’s Experimental Station, Mielżynskich Street 14, 64-122 Pawłowice, Poland; marikamy@wp.pl; 3Department of Animal Genetics, Breeding and Ethology, University of Agriculture, Mickiewicza 24/28, 30-059 Kraków, Poland; magdalena.pieszka@urk.edu.pl

**Keywords:** piglet rearing, pancreatic-like enzymes (PLEMs), IUGR

## Abstract

**Simple Summary:**

This study was carried out to evaluate the effect of enzymes of microbiological origin with pancreatic profiles added to the feed of pregnant sows on fetal size in order to minimize IUGR caused by fetal malnutrition. IUGR, understood as impaired growth and development of mammalian feti or their organs during pregnancy, is a serious problem in pig production. Restriction of fetal growth reduces the piglet survival rate, inhibits postnatal development, permanently weakens health, and also reduces pig production performance, negatively affecting meat quality. The research hypothesis was that the administration of PLEMs to sows would increase the uptake of nutrients by the feti, contributing to an increase in the piglets’ birth weight and survival rate, ensuring the proper growth and development of pigs. The development of effective strategies to reduce the incidence of IUGR and its effects will therefore improve the profitability of pig production. In conclusion, we can conclude that the use of fungal enzymes with pancreatic profiles has beneficial effects on piglets’ rearing rates, the development of their digestive tracts, and the activity of brush edge enzymes. It can be further concluded that the IUGR syndrome can be induced in pigs via the mother’s diet during pregnancy.

**Abstract:**

The present study aimed to develop a feeding strategy for pregnant sows that involved the prenatal administration of a mixture of pancreatic-like fungal enzymes, such as lipase, amylase, and protease, at (1) 1–115 days of gestation (group D1) and (2) 80–115 days of gestation (group D2) and to carry out a comparison with groups of sows that were not receiving such supplementation (negative control (NC) and positive control (PC)). It was found that the administration of the enzyme supplement resulted in a significant shortening of gestation (*p* ≤ 0.01). The pancreatic enzymes administered to sows had a significant effect on the number of liveborn piglets and weaned piglets, which was higher compared with the control groups that did not receive supplementation: D1—12.1 ± 1.1 and 11.12 ± 1.1 and D2—12.8 ± 1.3 and 11.75 ± 0.07 vs. the control groups KN—10.7 ± 1.0 and 9.62 ± 0.95 and KP—10.9 ± 1.2 and 10.15 ± 1.0 (*p* < 0.006), respectively. Significant changes in piglet growth were observed after weaning up to 70 days of age. During this period, the most favorable growth parameters were observed in groups D2 (420 ± 91 g) and PC (407 ± 103 g), in which piglets obtained a mixture of pancreatic enzymes (lipase, amylase, and protease) at 3 weeks of age, and significantly higher weight gain and feed intake were observed compared with groups NC (378 ± 114 g) and D1 (381 ± 96 g) (*p* ≤ 0.007). In contrast, insulin levels were significantly lower in groups D1 and D2, with values of 6.8 IU/mL and 6.7 IU/mL, respectively, compared with groups NC (14.6 IU/mL) and PC (16.6 IU/mL) (*p* ≤ 0.01). Piglets in group D2 had a significantly better feed conversion ratio (FCR) of 1.604 ± 0.10 compared with the other dietary groups: KN—1.986 ± 0.14; KP—1.704 ± 0.11; and D1—1.932 ± 0.15 (*p* ≤ 0.03). Histological imaging confirmed a significantly thicker intestinal epithelium and intestinal mesenteron in animals from groups D2 and PC (*p* ≤ 0.03). Animals from the groups KP, D1, and D2 receiving enzymes showed a highly significant increase in the surface area of pancreatic follicles and pancreatic surface area compared with the group without KN supplementation (*p* < 0.01). Furthermore, significantly higher activity of the brush border enzyme lactase was observed in groups D1, D2, and PC, with values of 32.90 ± 3.99, 30.00 ± 6.83, and 29.60 ± 29.60, respectively, compared with group NC, with a value of 21.80 ± 3.27 (*p* ≤ 0.01).

## 1. Introduction

Despite continuous improvements in herd management and intensive research into mammalian nutritional requirements with the use of modern analytical techniques, the intrauterine growth restriction (IUGR) syndrome remains a challenge in animal husbandry due to the unavailability of adequate literature on the effect of nutrition on feti growth regulation mechanisms [1]. Increasing the number of fetuses in the uterus without expanding the uterine volume results in relative placental failure and a low birth weight of the newborns [2]. Similar to humans, the IUGR syndrome arises spontaneously in piglets, affecting 6–10% of newborns, and increases in proportion to the increasing number of piglets in a litter. This high incidence of the syndrome has increasing economic consequences in livestock production [3]. Animals with the IUGR syndrome are characterized by a high mortality rate of approximately 85% in the first 2–3 days of postnatal life [4].

In livestock, the primary cause of the IUGR syndrome is an imbalanced diet. Interestingly, both a diet that is low in protein and a diet containing an excess of protein can lead to the development of the IUGR syndrome [5]. In rats and mice, the use of a high-protein diet in the second half of pregnancy has been shown to cause a significant reduction in neonatal birth weight [6]. These findings are in agreement with the growth slowdown observed during the last trimester of pregnancy in women who prefer a high-protein diet [7]. The main reason for this observation is a delay in blastocyst development caused by an excessive amount of toxic protein breakdown products circulating in the body of the mother [8]. A high-protein diet is also associated with an increase in the energy cost of urine production and hepatic gluconeogenesis, which can reduce the supply of energy components to the fetus. Earlier studies on pregnant female rodents and sheep showed that a high protein intake and an increase in the ammonia concentration (up to 300% higher compared with the control group) result in a reduction in the number of developing blastocysts, impaired embryo metabolism, and fetal growth disorders [9]. In pregnant mothers who receive a high-protein diet, the amino acid profile of the plasma is altered, and the concentrations of threonine, glutamine, glycine, alanine, and serine decrease, which slow down the growth of fetal tissues [10]. In mice, a high-protein diet had an impact on the GH/JAK/STAT/IGF signaling pathway, which was reflected in decreased growth hormone concentrations in the placenta and a consequent decrease in fetal weight gain [11]. The changes in lipid metabolism lead to an increase in the levels of annexin IV, a protein responsible for adipocyte differentiation and adipose tissue growth [12]. This is manifested in an increased predisposition to obesity in adults. Furthermore, an indirect consequence of a high-protein diet during pregnancy, which can have an effect even in the adulthood of individuals born with the IUGR syndrome, is changes in the cardiovascular system, leading to the development of cardiovascular diseases in humans [13] at the postnatal stages. Studies have also shown that in pregnant mice fed a high-protein diet, the plasma concentrations of non-esterified fatty acids increase to levels recorded in the blood of 24 h fasted animals. This indicates an increase in lipolysis and fat oxidation to compensate for the energy deficit in the developing fetus [14]. Similar to a high-protein diet, consumption of a diet with reduced protein during pregnancy results in a decrease in the birth weight of the newborn [15]. A reduced dietary protein supply leads to a decrease in the pool of amino acids in the maternal plasma and, consequently, in the fetal plasma and uterine fluid. This, in turn, translates into stunted fetal growth [16]. Studies on pigs have shown that a reduced dietary supply of amino acids decreases the efficiency of the amino acid transport protein system in the placenta [16]. In the placenta, the amino acid transport mechanisms are mainly based on the active transport of protein and require an energy supply to overcome the concentration gradient. In pregnant rats on a low-protein diet, there was a decrease in the plasma concentrations of the regulators of fetal development, such as insulin, leptin, low-density lipoprotein (LDL), and insulin-like growth factor 2 (IGF-2), which reduced the expression of the sodium-coupled neutral amino acid transporter 2 (SNAT2) protein involved in the placental transport of amino acids to the fetus [14]. Despite the restricted nutrient supply, increased fat accumulation by the fetus was observed, which was attributed to an increase in the expression of annexin V, a protein that stimulates adipocyte proliferation and differentiation [12]. Research on the effects of maternal diet imbalance during pregnancy and/or the influence of other factors relevant in humans, such as smoking, alcohol consumption, and drug intake, gave rise to the “thrifty phenotype” hypothesis. The thrifty phenotype is characterized not only by a low birth weight but also by a number of changes in the structures and functions of the internal organs, which are caused by an inadequate supply of glucose and amino acids to the fetus. These changes include alterations in the funcion of the kidneys, endocrine and exocrine parts of the pancreas, intestines, skeletal muscles, liver, and nervous system. Some of these changes may level off over time, such as the changes in the function of brush border enzymes in the intestinal mucosa, while some may not, such as the changes in the morphology of the kidneys (the number of nephrons) and pancreas (the number and size of pancreatic islets). Later stages of life and adulthood may be characterized by further effects associated with the IUGR syndrome, such as hypertension, type 2 diabetes, and other symptoms that are typical of the metabolic syndrome [17]. Other diseases observed in adults with the IUGR syndrome include visceral obesity and cardiovascular diseases as a consequence of hyperlipidemia and hypertension [13].

Proper sow nutrition is an important factor affecting the course of pregnancy and lactation and thus the growth, development, and survival of piglets. When a sow becomes pregnant, malnutrition from the preceding time period, combined with the limited feed intake of the gestating sow, negatively affects the growth and development of the early-stage embryos and feti [18]. Both malnutrition and the overfeeding of gestating sows result in a delayed growth of the feti [19].

An excessively high energy and/or protein content in the mixture given to a female after insemination and during early pregnancy increases embryonic and fetal mortality. Mostly, the IUGR syndrome occurs naturally in pigs due to the high fertility of the species. In advanced pregnancy, uterine capacity becomes a limiting factor for fetal growth. The development of feti depends on their location and number; the ones placed at the ends of the uterine horns are larger than those placed in the center due to the difference in blood pressure and nutritional status [20]. This fetal weight difference is magnified during late pregnancy when the number of feti per horn exceeds five [21]. At birth, the body weight of a piglet with the IUGR syndrome may range from only half to one-third of the body weight of the largest piglet in the litter [22]. The changes in the number and size of primary and secondary muscle fibers and the proportion of adipocytes prenatal during the period affect the quality of raw slaughter material and pork in the postnatal period and after the animals are fattened [23]. This has been confirmed by a national study conducted by Rekiel et al. [24], which determined the relationship between the weight of piglets at birth and the quality of raw slaughter material and pork. Thus, understanding the mechanisms behind the development of the IUGR syndrome and developing effective strategies to reduce the intensity of the syndrome as well as its consequences will help to improve the profitability of pig production. Currently, several factors are cited as causes of the IUGR syndrome, but the course of its development has not yet been sufficiently explained.

The hypothesis of this study is that the administration of fungal pancreatic-like enzymes of microbial origin (PLEMs) to sows during pregnancy will contribute to a significant improvement in placental function in terms of nutrient permeability to the fetus and in fetal development and growth while reducing the probability of the IUGR syndrome.

## 2. Materials and Methods

### 2.1. Research Scheme, Animals, and Feeding

The experiment was carried out on 24 Polish Landrace sows with their offspring. The sows were third-time pregnant and were chosen out of a group consisting of 900 sows. The sows were randomly divided into four groups, with six animals in each group, with a minimum of 12 liveborn piglets. The animals were fed with compound feed according to the nutrient requirements according to DLG standards [25] as shown in Table 1. Two control groups of 6 animals each (*n* = 6) were fed with a standard feed for farrowing sows without enzyme additives (Table 2). The difference between the two control groups (positive control (PC) and negative control (NC)) was that the piglets in one group (PC) received PLEM supplementation between 14 and 21 days of age. In addition to the above-mentioned groups, two experimental groups were also set up, each with six sows. One experimental group (group D1) received mixed feed for pregnant sows supplemented with microbial pancreatic enzymes throughout the pregnancies, while the other (group D2) received enzyme supplementation only in the last 4 weeks from day 80 of gestation (Table 2). Pregnant sows were maintained in individual farrowing pens, and piglets had constant access to the mothers. The sows were maintained under standard conditions with a controlled daylight length (7.00–7.00 p.m.) and temperature (22 ± 2 °C). After parturition, the piglets were reheated with radiant heaters to maintain the temperature at approximately 30 ± 2 °C. Piglets after weaning from sows were kept in separate pens by litters.

Throughout this study, the production cycle of sows and piglets was monitored. To evaluate the long-term effects of enzymes in sow feed on the body development of piglets from each litter, two piglets were euthanized immediately after birth, and biological material was collected for structural (gastrointestinal tract including the stomach, pancreas, duodenum, small intestine, and colon) and functional (concentrations of intestinal hormones, namely, ghrelin, leptin, and insulin, and activity of brush border enzymes) analyses. The pigs were sedated using azaperone (Stresnil, LEO, Helsingborg, Sweden) (5 mg/kg body weight) and euthanized using a single dose of IV-injected sodium pentobarbiturate (100 mg/kg body weight). Piglets were weaned after a 4-week lactation period, and the average weaning age was 28 days. After weaning, until 70 days of age, the piglets were reared in the piggeries in group pens with litters. From day 7 onward, all piglets were fed a standard super pre-starter mixture (Table 3). A weaning pre-starter mix was used until day 14. From day 15, the piglets were fed a starter mixture for the next 28 days until 70 days of age (Table 3). Both the weaning pre-starter mixture and the starter mixture were fed ad libitum. The chemical composition of the compound feed was analyzed according to AOAC [26] methods.

The enzymes (Amano Enzyme, Nagoya, Japan) used in this study were of microbial origin. Amylase with an activity of 90,000 DU/g was obtained from the fermentation of the fungus Aspergillus oryzae, protease with an activity of 150,000 HUT/g from the fermentation of the fungus Aspergillus melleus, and lipase with an activity of 15,000 U/g from the fermentation of the fungus Aspergillus niger.

The data on piglet rearing were collected during the experiment. These included body weight, daily gains, feed intake, and feed utilization, as well as the number of falls and the health status of the piglets. The ultrasonograph Aloka SSD-500 (Hitachi, Tokyo, Japan), equipped with a 17 cm UST-5044 head with an operating frequency of 3.5 MHz, was used to evaluate the degree of fattening of sows. Measurements were taken at 90 days of gestation at point P2. 

### 2.2. Analysis of Brush Border Enzyme Activity

The determination of saccharase, lactase, and maltase activities in the brush border was carried out as follows.

The activities of saccharase, lactase, and maltase were determined in the brush border of the jejunum using a modified method of Dahlquist [27]. Briefly, the intestinal epithelium samples collected at postmortem were weighed, diluted in distilled water (1:4 *v*/*v*), and thoroughly homogenized using a mechanical homogenizer. The resulting homogenates were placed on ice. Then, these samples were mixed with a solution containing substrate (sucrose, lactose, or maltose, respectively) and a reaction buffer with maleic acid. All the samples were incubated at 37 °C for 1 h, after which the activity of disaccharidases was stopped immediately by adding an inhibitory solution containing Tris. At the same time, blank samples were prepared, in which an inhibition solution was immediately added to the mixture of homogenates, substrate, and reaction buffer. The prepared samples were transferred to a microplate, and then commercially available Glucose RTU reagent was added to determine glucose concentrations. A glucose standard was prepared by mixing it with the inhibition solution, and a standard curve was plotted. The microplate containing the samples was incubated for 15 min at 37 °C, and the absorbance was read at 490 nm using a spectrophotometer. To convert the enzymatic activity of the disaccharidases, the total protein content of the homogenates was determined using a commercial assay and spectrophotometric method, following the manufacturer’s instructions.

### 2.3. Determination of the Activities of Aminopeptidases A and N and Dipeptidylpeptidase IV in the Brush Border

The activities of aminopeptidases A and N and dipeptidylpeptidase IV were determined in the brush border of the jejunum using a modified method of Maroux et al. [28]. Briefly, the intestinal epithelium collected at postmortem was weighed, diluted in distilled water (1:4 *v*/*v*), and thoroughly homogenized using a mechanical homogenizer. The homogenates were placed on ice. The enzyme activity was measured via spectrophotometry using the following synthetic substrates: l-glutamic acid p-nitroanilide, leucine p-nitroanilide, and glycyl-l-prolyl *p*-nitroanilide tosylate for aminopeptidase A, aminopeptidase N, and dipeptidylpeptidase IV, respectively. Each homogenate was mixed with the appropriate substrate and reaction buffer, and the reactions were carried out in cuvettes at 37 °C. The concentration of the final reaction product, para-nitroaniline, was determined by measuring the kinetic absorbance at 410 nm. A solution of the substrate for the corresponding enzyme was used as a blank. To convert the enzyme activity, the total protein contents of the homogenates were determined using a commercial assay and spectrophotometric method, following the manufacturer’s instructions.

### 2.4. Analysis of Hormones in Blood Plasma via Radioimmunoassay (RIA)

To determine the brush border enzyme activity, the epithelium was collected after the animals were euthanized and the digestive tract was dissected. After removing the digestive contents from the lumen of the small intestine, the mucosal layer was collected, placed immediately in dry ice, and transported to the laboratory. The samples were refrigerated at −80 °C until the analysis. For the analyses, blood was collected from six piglets in each feeding group on an empty stomach from the icteric vein into EDTA-treated tubes. The collected blood samples were slightly cooled and then centrifuged for 10 min at 3000 rpm. The plasma hormone concentrations were determined using commercial RIA kits. Leptin and insulin were determined in porcine plasma with an animal-species-dedicated kit (Merck Millipore Inc., Burlington, MA, USA), while ghrelin was assayed with a commercial porcine Ghrelin RIA kit (Phoenix Pharmaceuticals Inc., Burlingame, CA, USA). Each test was carried out by applying an appropriate methodology for the determination of the respective compound. All determinations were performed according to the accompanying protocol for each test. The commercial RIA test kit included tubes coated with a high-affinity antibody to the compound under study. The kit also included calibrators, standards, the J-125 isotope, antibodies, and a wash solution. 

During the incubation of standards, controls, and test samples, a number of hormone molecules labeled with the iodine isotope J-125 competed with the test hormone for a certain number of antihormone antibody-binding sites that were bound to the antibodies immobilized on the wall of the tube. After incubation (the incubation time and temperature depend on the type of compound under study), the reaction was stopped via aspiration. Then, the tubes were rinsed with the working rinse solution, aspirated, and placed in an LKB Wallac 1275 Miniggama ionizing radiation reader. For each tube, the radiation reading was taken for 60 s. Based on the radioactivity measurement of the radioactive isotope, a calibration curve was plotted, and the hormone concentration in the sample was determined by interpolating the dose from the calibration curve.

### 2.5. Gastrointestinal Tract Histology

For the histometry studies, gastrointestinal tissue samples were routinely dehydrated in a graded alcohol series, embedded in paraffin, cut into 4.5 μm sections using a rotary microtome (Microm 350, Wetzlar, Germany), placed on silane-treated glass slides, and then stained with hematoxylin and eosin. For the pancreas, the surface area of pancreatic alveoli and cells and the number of follicular cells per pancreatic vesicle were determined. For the stomach, the mucosa thickness and muscle membrane thickness were determined. For the intestinal histometry, the intestinal villi length, crypt depth, mucosa thickness, and thickness of the muscle membrane were determined. For each analysis, a minimum of 15 intact slides from each individual were assessed. The analyses were performed using a light microscope (Axioskop 40, Zeiss, Jena, Germany) coupled with computer software for image analysis (Axio Vision 4.2 Release, Zeiss, Jena, Germany) and a digital camera.

### 2.6. Statistical Analyses

Statistical analyses of the results of the piglet rearing analysis, histological analysis, and enzyme and hormone analyses were performed using one-way analysis of variance. To evaluate the differences between groups, multiple comparisons were conducted using Student’s *t*-test. Pens were used as experimental units for the analysis of ADFI and F:G, and for other parameters, each pig was used as the experimental unit. Statistical significance was assumed at *p* ≤ 0.05. All calculations were performed using the Statgraphics Plus 6.0 (2001) statistical package. 

## 3. Results

During the lactation period, the sows in groups D1 and D2 consumed more feed compared with the NC and PC groups, detailed as follows: 6.6 (±0.7) and 6.8 (±0.9) kg vs. 6.2 (±0.8) and 6.3 (±0.9) kg, respectively (Table 4). The mean weight of sows on the 90th day of pregnancy in the NC and PC control groups was significantly lower compared with the D1 group (*p* ≤ 0.05), with values of 215 (±3.20), 218 (±3.50), and 230 (±2.60), respectively, and in the D1 group, it was 236 (±2.40) kg (*p* ≤ 0.05). The average fat thicknesses measured on the back at point P2 on the 90th day of pregnancy in the NC, PC, D1, and D2 groups were 21.30 (±1.08), 21.45 (±1.04), 24.60 (±1.12), and 22.10 (±1.05) mm (*p* = 0.06), respectively. The weight losses of sows from farrowing to weaning were as follows: in group NC, 28.8 (±2.8) kg; in group PC, 29.2 (±3.1) kg; in group D1, 32.8 (±3.2) kg; and in group D2, 33.5 (±3.3) kg. Litters from the sows in both the NC and PC groups were characterized by lower numbers of liveborn piglets compared with those from the sows in the experimental groups, amounting to 10.7 (±1.0), 10.9 (±1.2), 12.1 (±1.1), and 12.8 (±1.3) (*p* ≤ 0.006) piglets, respectively, while on the day of weaning, the litter counts were as follows: in group NC, 9.62 (±0.95); in group PC, 10.15 (±1.0); in group D1, 11.12 (±1.0); and in group D2, 11.75 (±1.2) (*p* ≤ 0.009)(Table 4). The average weight of a piglet on the day of birth varied significantly between the control and experimental groups as follows: 1651 (±366) g in group NC and 1666 (±317) kg in group PC vs. 1731 (±308) g in group D1 and 1756 (±362) g in group D2 (*p* = 0.0214). The enzyme additions administered to sows during pregnancy resulted in a reduction in the number of stillborn piglets from 0.545 (±0.06) and 0.550 (±0.07) in the NC and PC groups to 0.289 (±0.04) and 0.335 (±0.03) in the experimental groups D1 and D2 (*p* ≤ 0.285) (Table 5). In the case of the number of mummified piglets, significantly lower numbers of such feti were found in sows from the experimental groups D1 and D2, amounting to 0.165 (±0.015) and 0.178 (±0.018), compared with the control groups NC and PC, amounting to 0.333 (±0.03) and 0.350 (±0.03) (*p* ≤ 0.012), respectively. The piglet losses during the entire piglet rearing period between 1 and 70 days of age were as follows: in group NC, 16.9%; in group PC, 6.9%; in group D1, 10.3%; and in group D2, 7.3%.

On the other hand, the average number of piglets born with the IUGR syndrome characterized by a reduced body weight of less than 1.1 kg was significantly different between group PC 0.98 (±0.09) and group D1 0.61 (±0.06), in which sows received enzymes throughout pregnancy (*p* ≤ 0.012) (Table 5).

The statistically significant variations in piglet weight between groups NC and PC and groups D1 and D2 persisted until the end of the period of rearing piglets with the mother, that is, until day 28 of life, and were as follows: in group NC, 6.79 (±1.54) kg; in group PC, 6.72 (±1.95) kg; in group D1, 6.92 (±1.43) kg; and in group D2, 7.52 kg (±1.48) (*p* ≤ 0.052). On day 14 after weaning, the weights of the piglets were as follows: in group NC, 8.90 (±2.05) kg; in group PC, 9.17 (±2.37) kg; in group D1, 9.39 (±1.55) kg; and in group D2, 9.61 (±1.71) kg (*p* = 0.0632). At the end of weaning, on day 70, the weights of piglets were as follows: in group NC, 19.50 (±4.84) kg; in group PC, 20.58 (±3.91) kg; in group D1, 20.06 (±3.60) kg; and in group D2, 21.67 (±3.68) kg. There were significant differences between group NC and group D2 (*p* ≤ 0.0384).

During the entire period of rearing piglets with the mother, from 1 to 28 days of age, a favorable effect of the additives on piglet weight gain was observed. In the experimental groups, the weight gain was as follows: in group D1, 185 g (±49) and in group D2, 205 g (±49) vs. in group NC, 183 g (±50) and in group PC, 180 g (±59) (*p* ≤ 0.1545). After weaning between day 28 and day 42, the piglet weight gain significantly varied between groups NC and PC and groups D1 and D2 as follows: 150 g (±67), 175 g (±75), 176 g (±58), and 171 g (±71) (*p* ≤ 0.0375), respectively. Between day 42 and day 70, the weight gains were as follows: in group NC, 378 g (±114); in group PC, 417 g (±103); in group E1, 381 g (±96); and in group D2, 420 g (±91) (*p* ≤ 0.0078). For the entire rearing period between 1 and 70 days of age, there were significantly higher piglet weight gains in groups D2 and PC compared with the control groups: in group D2, 284 g (±49); in group PC, 270 g (±56); in group NC, 255 g (±66); and in group D1, 262 g (±51) (*p* ≤ 0.0362). The average feed intake of piglets during the rearing period with the sow was similar in all piglet groups and ranged from 28g (±4) in group NC to 29 (±5), 34 (±4), and 33 (±5) g in groups PC, D1, and D2, respectively (Table 6). After the piglets were weaned on day 28, a limited feed intake of piglets was observed: in group NC, 216 (±16) g; in group PC, 235 (±19) g; in group D1, 218 (±17) g; and in group D2, 238 (±20) g (*p* ≤ 0.052). An increase in feed intake occurred after 42 days of age and between day 42 and day 70 in groups D2 and PC, while the feed intake was significantly lower in groups NC and D1 (*p* ≤ 0.028). For the entire rearing period between 1 and 70 days of age, a significantly higher feed intake was found in group D2 (412 g; ± 32) and control group PC (392 g; ± 28) compared with control group NC (338 g; ± 24) and group D1 (344 g; ± 28) (*p* ≤ 0.038). The index of the feed conversion rate in the first period of a piglet’s life was similar in all groups and ranged between 0.09 and 0.16 kg/kg. For the piglet rearing period between day 28 and day 42, the feed conversion ratio (FCR) index was 1.986 (±0.012) kg/kg in group NC and 1.932 (±0.014) kg/kg in group D1 and differed significantly compared with group D2, with a value of 1.604 (±0.011) kg/kg), and group PC, with a value of 1.704 (±0.013) kg/kg) (*p* ≤ 0.0308). For the entire rearing period from day 1 to day 70, the FCR indexes in groups NC, PC, D1, and D2 were 1.26 (±0.14), 1.18 (±0.11), 1.27 (±0.15), and 1.16 (±0.10) (*p* = 0.1016), respectively (Table 7). 

Histological analyses and the determination of brush border enzyme activity were performed on samples from six animals representing each nutritional group. The histological analysis of selected sections of the gastrointestinal tract did not reveal any pathological changes or inflammation in the analyzed tissues (Table 8). No lysosomal vacuoles were observed in any section of the small intestine or the hindgut. The results of the histological analysis of the pancreas are shown in Table 8. A highly significant increase in the pancreatic follicle area and pancreatic cell area was noticed in groups D1 and D2 and the control group PC compared with the control group NC (*p* ≤ 0.01). The number of pancreatic follicle cells was significantly lower number in group D1 compared with group NC (*p* ≤ 0.05).

The results of the histometric analysis of the stomach are shown in Table 9. The mucosa thickness was significantly lower in groups NC and D1 compared with groups PC and D2 (*p* ≤ 0.01). Similar results were observed for mucosa membrane thickness, with significant differences in groups NC and D1 compared with group D2 (*p* ≤ 0.05).

Table 10 shows the results of the histometric analysis of the duodenal mucosa. Dietary supplementation of sows with pancreatic enzymes did not affect the length of intestinal villi of the duodenum but caused a significant increase in the depth of intestinal crypts and thickness of the mucosa in animals from group PC in which piglets received enzymes at 3 weeks of age and group D2, compared with animals from groups NC and D1 (*p* ≤ 0.01). In the case of duodenal muscle membrane thickness, a significantly higher membrane thickness was found in groups PC and D2 compared with groups NC and D1(*p* ≤ 0.05).

As in the case of the duodenum, and also in the initial section of the small intestine, the length of the intestinal villi was not influenced by the additives fed to the sows (Table 11). Similarly, the diet had no effect on the depth of intestinal crypts. However, statistically significant differences were found in the mucosa and muscle membrane thicknesses, with significantly higher values in groups PC and D2 compared with groups NC and D1 (*p* ≤ 0.01). The feed additives used did not affect the length of intestinal villi and the depth of intestinal crypts in the middle section of the intestine in all the analyzed nutritional groups (Table 11). A highly significant increase in the mucosa and muscle membrane thicknesses was noted in the animals of groups D2 and PC compared with those of groups NC and D1 (*p* ≤ 0.01). The results of the morphometric analyses of the terminal intestine revealed that the addition of PLEMs to the diet fed to sows throughout the gestation period and to piglets in group PC led to highly significant increases in the depth of crypts and mucosa and muscle membrane thicknesses compared with groups NC and D1 (*p* ≤ 0.01). The results of the histometric analysis of the iliac colon wall structure are shown in Table 12. Microscopic analysis showed no differences in the length of intestinal villi between the study groups. Highly significant increases in the depth of intestinal crypts and mucosa and muscle membrane thicknesses were found in groups D2 and PC compared with groups NC and D1 (*p* ≤ 0.01).

In the proteins of the epithelium in the initial (prox.), middle (mid.), and distal (dist.) parts of the jejunum, the activities of the brush border enzymes, saccharase, lactase, maltase, aminopeptidase A and N, and dipeptidylpeptidase IV, were determined (Table 13). Significantly higher lactase activity in the initial part of the jejunum was observed in the piglets of groups D1, D2, and PC compared with group NC, with values of 32.9, 32.00, and 29.60 vs. 21.80 nM/min/mg protein (*p* ≤ 0.01), respectively. In the case of the other enzymes, no significant differences were noted between the groups.

In the blood plasma of euthanized piglets, the indicators of immunity and body inflammation were determined. The results show that the level of immunoglobulin IgA was the same in all nutritional groups (Table 14). A significantly higher level of acute-phase protein haptoglobin was found in group D1 and group NC compared with groups PC and D2 (*p* ≤ 0.01) (Table 14).

The plasma contents of hormones responsible for lipid and sugar metabolism were determined. A significantly higher plasma level of ghrelin was noted in the piglets of group NC compared with those of group D1, with values of 765 vs. 448 pg/mL (*p* ≤ 0.01) (Figure 1).

Regarding the plasma leptin content, no significant differences were found between the experimental groups and the values determined in groups NC, PC, D1, and D2 were 5.10, 4.05, 5.31, and 6.01 ng/mL, respectively (Figure 2).

The plasma insulin contents in piglets from sows in groups D1 and D2 were significantly lower compared with those in groups NC and PC, with values of 6.8 and 6.7 IU/mL vs. 14.6 and 16.6 IU/mL, respectively (*p* ≤ 0.01) (Figure 3).

## 4. Discussion

The IUGR syndrome is a condition characterized by the disruption of fetal development by nongenetic factors, mainly environmental ones, in the second and/or third trimester of pregnancy. Newborns with this syndrome are born at term with a low birth weight of approximately 1.1 kg, which is taken as the cutoff value for bacon breeds of pigs. A serious economic challenge for pig farms is that individuals born with the IUGR syndrome have a much higher mortality rate compared with normal piglets [29]. In Europe and Asia, where large numbers of pigs are raised, the average mortality rate among live piglets before weaning—depending on the source—is 11–13%, while the percentage of piglets born dead is 7–8% [30]. The main causes of death of newborn piglets are low body weight (14%), starvation (7%), crushing by the mother (5%), and diarrhea (4%), while limited milk intake is the most significant cause of all deaths. Piglets with limited or no milk intake are at risk of starvation, crushing, or diarrhea. Piglet survival is associated with complex interactions between the sow, the piglet, and their environment. In fact, it has been reported that in the European Union countries, 20% of liveborn piglets die before weaning and the riskiest group is piglets with a low birth weight. Approximately 50% of piglet losses can be attributed to hypothermia and crush, that is, factors primarily related to the piglets’ poor conditions/apathy, their ability to escape from a moving sow, or competition for access to colostrum/milk during the first days of life. Genetic selection strategies have increased litter size and practically reduced the birth/death rates of pigs [31]. The results of this study indicate the positive effect of exogenous pancreatic enzymes administered to pregnant sows, which was reflected in a higher number of liveborn piglets and better piglet survival. These piglets were characterized by, among others, a higher weight gain, feed intake, and final body weight at 70 days of age. All these favorable results could have been due to better fetal nutrition during the prenatal period and balanced body weight.

Studies conducted on sows giving birth to a large number of low-weight piglets have shown that mortality in the first 2–3 days of postnatal life can be as high as 85% [30]. This relationship between litter size and piglet mortality can be attributed to a number of factors, such as prolonged farrowing and associated hypoxia in the offspring and increased competition for access to the nipples and nutrients in the milk after birth [31]. The major factor predisposing piglets to increased mortality in the litters of hyperproliferative sows is the reduced viability of newborn piglets, low motility of piglets (failure to suckle, crushes by the mother, chronic stress, and subsequent high susceptibility to diseases, mainly those affecting the gastrointestinal tract, such as diarrhea), and an ill-balanced diet in terms of the amount of protein and/or energy consumed [32]. In intensive swine production farms, the incidence of the IUGR syndrome is at 6–10% of all births [33]. Several causes of the IUGR syndrome have been identified in recent years, but the course of development of this syndrome has not yet been sufficiently elucidated. There are many papers describing the etiology and underlying mechanisms of IUGR, which are related to deficiencies in both embryo quality and uterine capacity [34]. The environmental factors associated with the IUGR syndrome include viral and bacterial infections, which can account for up to 40% of cases. The present study revealed a positive effect of the administration of microbial pancreatic enzymes to sows during gestation, which became apparent in piglets, especially during the weaning period. This outcome was maintained until the end of the experiment (day 70), as manifested in higher gains and higher piglet survival, which may indicate the stimulating and long-term effect of the enzyme mixture. A number of recent studies have shown the close relationship between fetal and neonatal nutrition in the early postnatal period and proper body function in adulthood [5]. The most obvious consequence of the IUGR syndrome is increased piglet mortality in the neonatal period. The relationship between birth weight and survival rate has been well established. It was shown that the preweaning survival rate progressively decreased from 95% to 15% as piglet birth weight decreased from 1.80 to 0.61 kg [33]. In addition, low-birth-weight piglets that survive consistently show lower postnatal growth rates. The long-term effects of IUGR on piglet welfare have not been investigated in detail to date. An important factor determining the productivity of sows is adequate nutrition, which affects the course of pregnancy and lactation, and thus the growth, development, and survival of offspring. Low feed intake by lactating sows during the lactation period preceding mating/insemination causes the mobilization of the body’s reserves for milk production and results in the overfeeding of pregnant sows, causing delayed fetal growth [19]. Administration of a feed mixture with an excessively high energy and/or protein content to the female after mating and during early pregnancy increases embryonic and fetal mortality. A study conducted by Bee [35] showed that an approximately 43% higher intake of protein and energy by day 50 of gestation (relative to the standard feeding level for pregnant multiparous sows) resulted in a reduction in neonatal body weight. The small intestine plays an important role in the final digestion and absorption of nutrients and thus in the postnatal growth of animals. Naturally occurring or experimentally induced IUGR is associated with an abnormal morphology of the small intestine, which worsens the animal’s utilization of nutrients [36] and disrupts skeletal muscle development [36]. Newborns with the IUGR syndrome often suffer from necrotizing enterocolitis. This condition impairs intestinal function, including the synthesis of arginine, an amino acid that is essential for newborns but deficient in sow’s milk. It is also one of the leading causes of neonatal death [37]. Compared with feti or piglets with optimal weight, those with the IUGR syndrome show a slower growth rate and higher contents of intramuscular fat and connective tissue, mainly collagen I [38].

The newborn abnormalities characterized by low birth weight do not only affect piglets. Every year, in Poland, approximately 6% of human infants are born with a low birth weight and other features characteristic of the IUGR syndrome. In addition, the IUGR syndrome has not only been described in the offspring of humans and pigs, but also in rodents, rabbits, calves, and lambs, which has important implications for breeding. The digestive system of piglets born with the IUGR syndrome is characterized by delayed development, lower weight, abnormalities in motility, and improper digestion and absorption of nutrients compared with animals born with normal birth weight. Piglets with the IUGR syndrome have significantly thinner mucosa and muscle membranes and a significantly increased proportion of fetal-type enterocytes in the mucosa. The present study showed that the administration of pancreatic-like fungal enzymes to sows at the end of gestation caused an increase in the thickness of the mucosa and muscle membranes of the intestine in piglets as well as an increase in the entire length of the small intestine and the hind intestine. Due to the sow’s inability to produce more of its own pancreatic enzymes, providing additional pancreatic enzymes with the diet may alter the quality of digestion. A few years ago, it was proven that the same protein digested with the same enzymes but in different proportions can be a source of very different absorbable peptides with significantly different biological properties [39]. This property appears to be the main reason for the development of larger feti. Simply put, feti are provided with the right amount of nutrients through the mother’s blood to ensure the proper and rapid development of the feti.

In a study on the dietary administration of pancreatic enzymes in suckling piglets, Slupecka et al. [40] found that the growth of the small intestine was stimulated by the enzymes, as manifested in the increased mitotic index of crypts, height of villi, and depth of crypts. In the same study, the authors observed a decrease in the mucosa membrane thickness and the shortening of villi and crypts as a result of direct dietary administration of microbial-derived enzymes (PLEMs), which may indicate the sensitivity of piglets to endogenous enzymes [40]. In the present study, in which enzymes were administered to sows during gestation, a significant variation in the length of enterocytes was observed in piglets; however, the cells were shaped differently in different sections, and hence, it is difficult to draw a clear conclusion regarding the effect of prenatal administration of exogenous pancreatic enzymes on the size of enterocytes. Feeding sows with the enzyme preparation had no inhibitory effect on the pancreas of the piglets and even stimulated the proliferation of pancreatic cells, which indicates that the enzyme preparation had a trophic effect on the organ. In a study performed by Mickiewicz et al. [5], a number of maturation abnormalities of the small intestinal mucosa were observed in piglets born with the IUGR syndrome. Similar findings were presented by Amdi et al. [4], suggesting that growth abnormalities noted in piglets with the IUGR syndrome may not be due to nutrient deficiency, but rather to impaired nutrient uptake and impaired transfer of biologically active factors from the intestinal lumen to the blood. The increased proportion of fetal-type enterocytes in the mucosa of piglets with the IUGR syndrome may be due to the inhibition of apoptosis within the villi peaks and the reduced number of mitotic divisions within the intestinal crypts. The results of the presented studies indicate modified mucosal development in piglets with the IUGR syndrome compared with piglets born with normal birth weight. According to the literature, the course of development of the gastrointestinal tract in individuals with the IUGR syndrome is different from that of individuals born with normal birth weight, and these changes may be responsible for, among others, the disorders in the closing of the intestinal barrier and absorption [41].

This study also investigated the effect of exogenous pancreatic enzyme administration on the level of brush border enzyme activity. Interestingly, in piglets from sows receiving the enzymes (groups D1, D2, and PC) during the rearing period with the sow, significantly higher lactase activity was found after weaning the piglets, which is consistent with the results of Marion et al. [42]; however, most authors have shown a decrease in the activity of this enzyme [43,44]. It can be speculated that the prenatal administration of enzymes contributed to an improvement in the development of the intestine, including the muscle and mucosa membranes of the duodenum and small intestine, sites where brush epithelial enzymes are synthesized after birth. In addition, the relatively high proportion of skimmed milk in the piglet mix (16%) may have affected the activity of lactase. On the other hand, no significant differences were found in the activity levels of the other examined epithelial enzymes.

Based on the above findings, it can be concluded that the effects of weaning may affect the activity of intestinal enzymes, which appear to be age-dependent during weaning. Such age-dependent differences in the activity of brush border enzymes may be due to changes in the rate of cellular renewal (crypt cell proliferation, epithelial cell migration from crypts, and villus cell apoptosis) and protein synthesis (gene expression, protein maturation, and stability) [43]. The activity of pancreatic and gastric enzymes may be sufficient in pigs, except for in the postweaning period. Studies clearly indicate that diet is the most important factor affecting the development of the digestive system, especially the reorganization of the small intestinal mucosa. Thus, administration of feed with an unfavorable composition to newborns may disrupt the development of the gastrointestinal tract and even the entire body [45].

The present study also analyzed the levels of selected hormones responsible for cellular metabolism. Ghrelin is a key hormone influencing appetite stimulation and is hence known as the “hunger hormone.” It is synthesized mainly in the fundus and body of the stomach by X/A neuroendocrine cells [46]. In addition, this hormone participates in the regulation of the body’s carbohydrate and lipid metabolism by reducing insulin secretion in the pancreas and enhancing the process of adipogenesis. The significant reduction in blood insulin levels correlates with lower levels of ghrelin, as observed in the present study on piglets from sows receiving exogenous pancreatic enzymes.

The presence of ghrelin in the perinatal period and its important physiological and endocrine functions indicate that this hormone may also play a role in the development of the gastrointestinal tract, and perhaps in its adaptations associated with the IUGR syndrome [36]. However, only a few studies have described the role of ghrelin in the development of the gastrointestinal tract in neonatal and suckling animals [47]. In the present study, reduced levels of ghrelin were observed in piglets from sows receiving exogenous pancreatic enzymes in the diet, while insulin levels were significantly increased, which is consistent with the results reported by Xiong et al. [48]. In a study conducted by Chen et al. [49], significantly lower plasma levels of adiponectin and leptin were observed in normal-weight piglets affected by the IUGR syndrome compared with normal-weight piglets without the syndrome. In the present study, no significant differences in leptin levels were found between the groups, which may be due to the fact that the average body weight of piglets was similar, and thus their metabolic levels were also similar.

## 5. Conclusions

Based on the results, it was observed that the formulation of pancreatic-like digestive enzymes administered to piglets allowed for an increase in the assimilation of nutrients by the fetus, resulted in weight-adjusted newborns, and had a stimulating effect on the growth and normal development of piglets. It can be further concluded that the IUGR syndrome can be induced in pigs by the mother’s diet during pregnancy. The use of a fungal PLEM supplement was effective in reducing the occurrence of this syndrome in piglets. Research on the development of the digestive system can contribute to improving the welfare of farm animals. The resulting knowledge may allow the development of optimal feeding strategies for both pre-weaned and weaned animals. Current methods of intensive pig breeding applied in large farms are often based on the introduction of artificial feeding programs using solid feed. However, it should be noted that the mucosa of the small intestine is not fully mature and prepared to digest and absorb this type of feed, and this leads to disturbances in digestive processes, resulting in intestinal inflammation, diarrhea, and animal falls.

## Figures and Tables

**Figure 1 animals-13-03448-f001:**
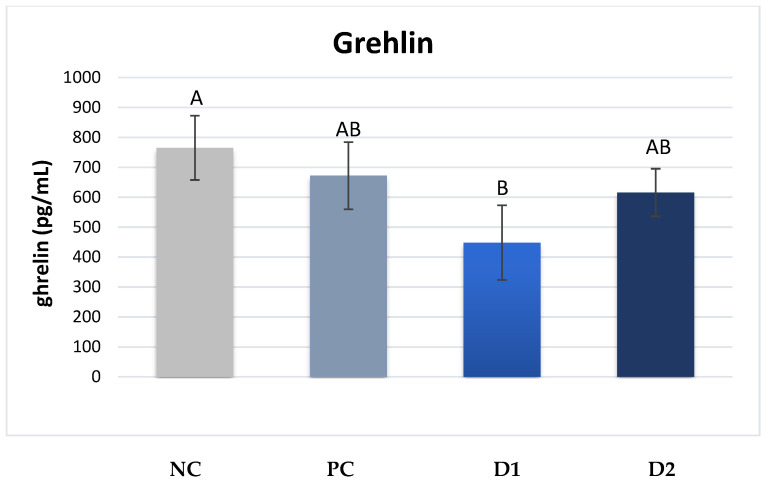
Plasma ghrelin levels in piglets. ^A,B^ Mean values denoted by the same letters are significantly different from each other at *p* ≤ 0.01.

**Figure 2 animals-13-03448-f002:**
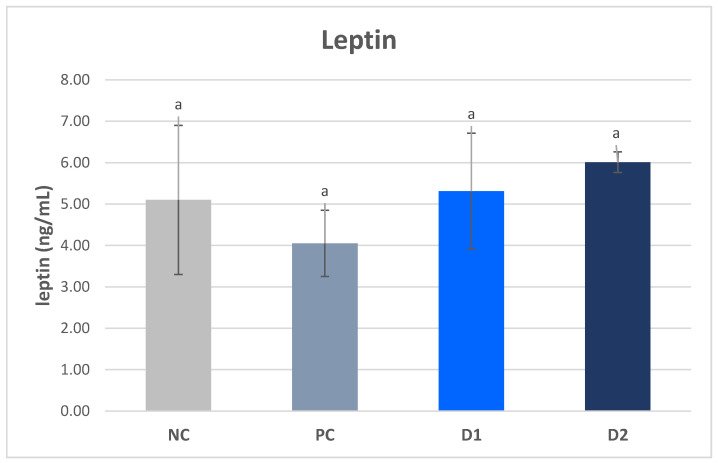
Plasma leptin levels in pigs. ^a^ Mean values denoted by the same letters are significantly different from each other at *p* ≤ 0.05.

**Figure 3 animals-13-03448-f003:**
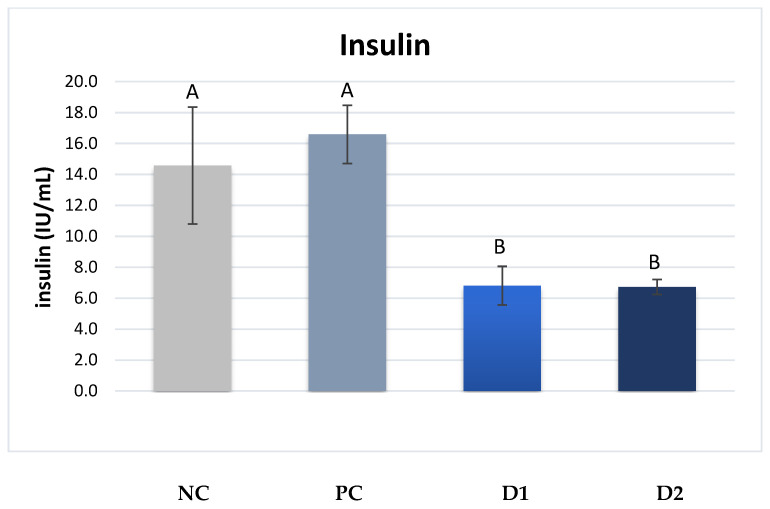
Plasma insulin contents in piglets. ^A,B^ Mean values denoted by the same letters are significantly different from each other at *p* ≤ 0.01.

**Table 1 animals-13-03448-t001:** Compositions and nutritional values of mixtures for sows.

Mixture Composition		Type of Feed Mixture	
	Unit	Sows in Early Gestation	Sows in Late Gestation
Barley meal	%	30.00	25.0
Oat meal	%	26.00	-
Triticale meal	%	19.00	20.00
Wheat meal	%	-	20.00
Wheat bran	%	15.69	-
Pszenmix	%	-	3.31
Dried grasses	%	2.00	-
Soybean meal	%	4.00	14.00
Forage chalk	%	1.00	0.70
Monocalcium phosphate	%	0.50	0.70
Feed salt	%	0.40	0.45
L-lysine	%	0.11	0.33
Mineral and vitamin premix	%	0.50	0.50
Rapeseed oil	%	0.50	0.50
Binder	%	0.30	-
		100.00	100.00
Nutrient content in 1 kg of DM			
Metabolic energy	MJ	11.5	13.1
Crude fiber	g	70	162
Crude protein	g	129	33
Lysine	g	6.0	9.4
Methionine + cystine	g	4.6	5.7
Threonine	g	4.2	6.1
Tryptophan	g	1.6	1.9
Calcium	g	6.0	7.5
Phosphorus	g	5.7	5.5
Sodium	g	1.9	2.0

DM: dry matter.

**Table 2 animals-13-03448-t002:** Enzyme doses administered to sows during the experiment.

Specification	Nutritional Group			
	NC	PC *	D1	D2
Administration period (days)	0	0	1–114	80–114
	Dose, IU/kg BW			
Lipase	0	0	1000	1000
Protease	0	0	1350	1350
Amylase	0	0	4000	4000

* Administration of microbial pancreatic enzymes (lipase: 10,000 IU/kg BW; protease: 650 IU/kg BW; and amylase: 6650 IU/kg BW) to piglets in the super pre-starter mixture between 14 and 21 days of age. BW: body weight.

**Table 3 animals-13-03448-t003:** Compositions and nutritional values of mixtures for piglets.

Mixture Composition	Unit	Type of Feed Mixture		
		Super Pre-Starter	Weaning Pre-Starter	Starter
Barley meal	%	13.06	40.00	40.00
Wheat meal	%	30.00	20.00	22.00
Corn meal	%	20.00	10.00	10.00
Soybean meal	%	5.00	-	17.00
HP 300	%	10.00	14.00	-
Skimmed milk powder	%	16.00	7.00	4.00
Pszenmix	%	-	3.55	4.00
Bergafat	%	-	2.00	1.00
Calcium formate	%	1.50	1.20	-
Phosphate 1-Ca	%	0.70	0.80	0.80
Feed salt	%	0.26	0.30	0.40
L-lysine	%	0.31	0.40	0.45
DL-methionine	%	0.02	0.05	0.08
L-threonine	%	0.07	0.13	0.14
L-tryptophan	%	0.03	0.03	-
Probiotic preparation	%	0.05	0.04	0.06
PP	%	0.50	0.50	0.50
Plant oil	%	2.50	-	0.5
		100.00	100.00	100.00
Nutrient content in 1 kg of DM				
Metabolic energy	MJ	13.9	13.5	13.2
Crude fiber	g	26	30	37
Crude protein	g	212	189	182
Lysine	g	13.5	12.5	11.9
Methionine + cystine	g	7.18	6.6	6.5
Threonine	g	8.52	8.0	7.6
Tryptophan	g	2.76	2.5	2.2
Calcium	g	8.66	7.6	7.6
Phosphorus	g	5.92	6.5	6.0
Sodium	g	1.50	1.6	2.0

DM: dry matter; PP: pre-starter premix.

**Table 4 animals-13-03448-t004:** Sow health indicators during piglet rearing.

Specification	Nutritional Group				*p*-Value
	NC	PC *	D1	D2	
Number of sows	6	6	6	6	
Mean weight of sows at 90 days of gestation (kg)	215 ± 3.20 ^a^	218 ± 3.50 ^a^	236 ± 2.40 ^b^	230 ± 3.70 ^ab^	0.05
Average fat thickness at P2 at 90 days of gestation (mm)	21.30 ± 1.08	21.45 ± 1.04	24.60 ± 1.12	22.10 ± 1.05	0.06
Length of pregnancy (days)	115.8 ± 2.4 ^b^	115.2 ± 2.0 ^b^	114.7 ± 2.2 ^a^	114.9 ± 2.2 ^a^	0.01
Average feed intake during lactation (kg)	6.2 ± 0.7	6.3 ± 0.9	6.6 ± 0.8	6.8 ± 0.9	≤0.122
Average number of piglets per litter	10.7 ± 1.0 ^a^	10.9 ± 1.2 ^a^	12.1 ± 1.1 ^b^	12.8 ± 1.3 ^b^	≤0.006
Average number of piglets with IUGR syndrome with ≤ 1.1 kg BW	0.86 ± 0.09 ^ab^	0.98 ± 0.11 ^b^	0.61 ± 0.06 ^a^	0.77 ± 0.07 ^ab^	≤0.012
Number of piglets after weaning	9.62 ± 0.95 ^a^	10.15 ± 1.0 ^a^	11.12 ± 1.1 ^b^	11.75 ± 1.2 ^b^	0.009
Average weight loss of the sows during the piglet feeding period (kg)	28.8 ± 2.8	29.2 ± 3.1	32.8 ± 3.2	33.5 ± 3.3	0.068
Number of cases of MMA (mastitis, metritis, and agalactia) syndrome in sows (heads)	1	0	0	0	
Piglet falls between 1 and 70 days of age (%)	16.9	6.9	10.3	7.3	

* Administration of microbial pancreatic enzymes (lipase: 10,000 IU/kg BW; protease: 650 IU/kg BW; and amylase: 6650 IU/kg BW) to piglets in the super pre-starter mixture between 14 and 21 days of age. ^a,b^ Mean values denoted by the same letters are significantly different from each other at *p* ≤ 0.05.

**Table 5 animals-13-03448-t005:** Piglet rearing indicators.

Specification	Nutritional Group				*p*-Value
	NC	PC	D1	D2	
Body weight at birth (g)	1651 ± 366	1666 ± 317	1731 ± 308	1756 ± 362	0.2143
Number of stillborn piglets	0.545 ± 0.06 ^b^	0.550 ± 07 ^b^	0.289 ± 04 ^a^	0.335 ± 03 ^a^	0.0285
Number of mummified piglets	0.333 ± 0.03 ^b^	0.350 ± 0.03 ^b^	0.165 ± 0.015 ^a^	0.178 ± 0.018 ^a^	0.016
Age at weaning (days)	27.4 ± 0.8 ^b^	27.2 ± 1.3 ^ab^	26.7 ± 0.8 ^a^	27.1 ± 1.7 ^ab^	0.0248
Body weight on weaning day or day 28 of life (kg)	6.79 ± 1.54	6.72 ± 1.95	6.92 ± 1.43	7.52 ± 1.48	0.052
Daily gain from day 1 to day 28 of life (g)	183 ± 50	180 ± 59	185 ± 49	205 ± 49	0.1545
Body weight on day 42 of life (kg)	8.90 ± 2.05	9.17 ± 2.37	9.39 ± 1.55	9.91 ± 1.71	0.0632
Daily gain from day 1 to day 42 of life (g)	173 ± 46	177 ± 50	182 ± 35	187 ± 37	0.0596
Daily gain from day 28 to day 42 of life (g)	150 ± 67 ^a^	175 ± 75 ^b^	176± 58 ^b^	171 ± 71 ^b^	0.0375
Body weight on day 70 of life (kg)	19.50 ± 4.84 ^ab^	20.58 ± 3.91 ^a^	20.06 ± 3.60 ^c^	21.67 ± 3.68 ^bc^	0.0384
Daily gain from day 1 to day 70 (kg)	255 ± 66 ^a^	270 ± 56 ^ab^	262 ± 51 ^a^	284 ± 49 ^b^	0.0362
Daily gain from day 42 to day 70 (kg)	378 ± 114 ^aA^	407 ± 103 ^bA^	381 ± 96 ^aA^	420 ± 91 ^cB^	0.0078

^a,b,c^ Mean values denoted by the same letters are significantly different from each other at *p* ≤ 0.05. ^A,B^ Mean values denoted by the same letters are significantly different from each other at *p* ≤ 0.01.

**Table 6 animals-13-03448-t006:** Average feed intakes of piglets during rearing (g/day/head).

Specification	Nutritional Group				*p*-Value
	NC	PC	D1	D2	
Rearing period, 1–28 days (g)	28 ± 4	33 ± 5	27 ± 4	34 ± 5	≤0.346
Rearing period, 28–42 days (g)	216 ± 16	235 ± 19	218 ± 17	238 ± 20	≤0.524
Rearing period, 1–70 days (g)	338 ± 24 ^ab^	392 ± 28 ^a^	344 ± 25 ^ab^	412 ± 32 ^b^	≤0.038

^a,b^ Mean values denoted by the same letters are significantly different from each other at *p* ≤ 0.05.

**Table 7 animals-13-03448-t007:** Feed utilization by piglets per kg of weight gain (kg/kg).

Specification	Nutritional Group				*p*-Value
	NC	PC	D1	D2	
FCR during the rearing period, 1–28 days	0.16 ± 0.02	0.10 ± 0.01	0.12 ± 0.01	0.09 ± 0.01	0.6362
FCR during the rearing period, 28–42 days	1.986 ± 0.14 ^a^	1.704 ± 0.11 ^ab^	1.932 ± 0.15 ^a^	1.604 ± 0.10 ^b^	0.0308
FCR during the rearing period, 1–70 days	1.26 ± 0.012	1.18 ± 0.013	1.27 ± 0.014	1.16 ± 0.011	0.1016

^a,b^ Mean values denoted by the same letters are significantly different from each other at *p* ≤ 0.05.

**Table 8 animals-13-03448-t008:** Morphometric analysis of piglets’ pancreases.

Specification	Nutritional Group			
	NC	PC	D1	D2
Surface area of pancreatic follicles (mm^2^)	692 ± 201 ^A^	788 ± 190 ^B^	890 ± 261 ^B^	855 ± 223 ^B^
Number of follicular cells per pancreatic follicle	9.4 ± 2.2 ^b^	8.5 ± 2.0 ^a^	8.1 ± 2.1 ^a^	8.7 ± 2.0 ^ab^
Pancreatic cell surface area (mm^2^)	72 ± 19 ^A^	84 ± 18 ^B^	89 ± 16 ^B^	87 ± 22 ^B^

Group NC: in this control group, sows did not receive pancreatic enzymes during gestation. Group PC: in this control group, the piglets received pancreatic enzymes between 14 and 21 days of age. Group D1: in this experimental group, sows received pancreatic enzymes between 80 days of gestation and farrowing. Group D2: in this experimental group, sows received pancreatic enzymes between 30 days of gestation and farrowing. ^A,B^ Mean values marked with different letters are statistically significantly different at *p* ≤ 0.01; ^a,b^ mean values marked with different letters are statistically significantly different at *p* ≤ 0.05.

**Table 9 animals-13-03448-t009:** Morphometric analysis of the mucosa and muscle membrane of piglet stomachs.

Specification	Nutritional Group			
	NC	PC	D1	D2
Mucosa thickness (µm)	589 ± 71 ^A^	655 ± 80 ^AB^	580 ± 68 ^A^	669 ± 85 ^B^
Muscle membrane thickness (µm)	1774 ± 227 ^a^	1852 ± 246 ^ab^	1800 ± 204 ^a^	1926 ± 258 ^b^

Group NC: in this control group, sows did not receive pancreatic enzymes during gestation. Group PC: in this control group, the piglets received pancreatic enzymes between 14 and 21 days of age. Group D1: in this experimental group, sows received pancreatic enzymes between 80 days of gestation and farrowing. Group D2: in this experimental group, sows received pancreatic enzymes between 30 days of gestation and farrowing. ^A,B^ Mean values marked with different letters are statistically significantly different at *p* ≤ 0.01; ^a,b^ mean values marked with different letters are statistically significantly different at *p* ≤ 0.05.

**Table 10 animals-13-03448-t010:** Histometric analysis of the mucosa and muscle membrane of the duodenums of piglets.

Specification	Nutritional Group			
	NC	PC	D1	D2
Length of villi (µm)	234 ± 38	238 ± 28	228 ± 31	226 ± 26
Depth of crypts (µm)	104 ± 12 ^A^	122 ± 16 ^B^	102 ± 14 ^A^	116 ± 12 ^AB^
Mucosa membrane thickness (µm)	483 ± 51 ^a^	506 ± 48 ^b^	488 ± 57 ^a^	497 ± 75 ^b^
Muscle membrane thickness (µm)	142 ± 21 ^A^	152 ± 20 ^B^	140 ± 22 ^A^	149 ± 18 ^B^

Group NC: in this control group, sows did not receive pancreatic enzymes during gestation. Group PC: in this control group, the piglets received pancreatic enzymes between 14 and 21 days of age. Group D1: in this experimental group, sows received pancreatic enzymes between 80 days of gestation and farrowing. Group D2: in this experimental group, sows received pancreatic enzymes between 30 days of gestation and farrowing. ^A,B^ Mean values marked with different letters are statistically significantly different at *p* ≤ 0.01; ^a,b^ mean values marked with different letters are statistically significantly different at *p* ≤ 0.05.

**Table 11 animals-13-03448-t011:** Histometric analysis of the mucosa and muscle membrane of different sections of the small intestines of piglets.

Specification	Nutritional Group			
	NC	PC	D1	D2
Initial section of small intestine				
Length of villi (µm)	272 ± 49	287 ± 33	277 ± 31	268 ± 32
Depth of crypts (µm)	101 ± 8	107 ± 8	108 ± 7	109 ± 7
Mucosa membrane thickness (µm)	540 ± 61 ^A^	596 ± 53 ^B^	555 ± 48 ^A^	591 ± 55 ^B^
Muscle membrane thickness (µm)	110 ± 15 ^A^	130 ± 20 ^BC^	117 ± 25 ^A^	139 ± 23 ^B^
Middle section of small intestine				
Length of villi (µm)	312 ± 35	320 ± 33	310 ± 34	321 ± 38
Depth of crypts (µm)	103 ± 4 ^aA^	118 ± 5 ^b^	106 ± 8 ^a^	122 ± 5 ^b^
Mucosa membrane thickness (µm)	416 ± 37 ^A^	434 ± 35 ^AB^	422 ± 38 ^A^	440 ± 39 ^B^
Muscle membrane thickness (µm)	112 ± 18 ^A^	124 ± 15 ^AB^	114 ± 19 ^A^	127 ± 20 ^B^
Terminal section of small intestine				
Length of villi (µm)	330 ± 36	334 ± 36	326 ± 33	333 ± 30
Depth of crypts (µm)	102 ± 5 ^A^	116 ± 5 ^AB^	107 ± 7 ^A^	124 ± 6 ^B^
Mucosa membrane thickness (µm)	442 ± 46 ^A^	472 ± 46 ^Ab^	450 ± 50 ^A^	482 ± 49 ^B^
Muscle membrane thickness (µm)	119 ± 22 ^A^	126 ± 22 ^AB^	122 ± 19 ^A^	128 ± 20 ^B^

Group NC: in this control group, sows did not receive pancreatic enzymes during gestation. Group PC: in this control group, the piglets received pancreatic enzymes between 14 and 21 days of age. Group D1: in this experimental group, sows received pancreatic enzymes between 80 days of gestation and farrowing. Group D2: in this experimental group, sows received pancreatic enzymes between 30 days of gestation and farrowing. ^A,B,C^ Mean values marked with different letters are statistically significantly different at *p* ≤ 0.01; ^a,b^ mean values marked with different letters are statistically significantly different at *p* ≤ 0.05.

**Table 12 animals-13-03448-t012:** Histometric analysis of the mucosa and muscle membranes of the piglets’ iliac colons.

Specification	Nutritional Group			
	NC	PC	D1	D2
Length of villi (µm)	483 ± 30	450 ± 34	440 ± 39	436 ± 38
Depth of crypts (µm)	106 ± 7 ^A^	118 ± 8 ^B^	104 ± 7 ^A^	116 ± 9 ^B^
Mucosa membrane thickness (µm)	487 ± 55 ^A^	526 ± 54 ^B^	490 ± 54 ^A^	516 ± 56 ^B^
Mucosa membrane thickness (µm)	113 ± 11 ^A^	134 ± 14 ^B^	114 ± 13 ^B^	126 ± 12 ^B^

Group NC: in this control group, sows did not receive pancreatic enzymes during gestation. Group PC: in this control group, the piglets received pancreatic enzymes between 14 and 21 days of age. Group D1: in this experimental group, sows received pancreatic enzymes between 80 days of gestation and farrowing. Group D2: in this experimental group, sows received pancreatic enzymes between 30 days of gestation and farrowing. ^A,B^ Mean values marked with different letters are statistically significantly different at *p* ≤ 0.01.

**Table 13 animals-13-03448-t013:** Activities of piglet jejunum brush border enzymes (nM/min/mg protein).

Enzyme	Jejunum Part	Nutritional Group			
		NC	PC	D1	D2
Saccharase	Prox.	0.50 ± 0.17	0.50 ± 0.13	0.68 ± 0.11	1.36 ± 046
	Mid.	0.68 ± 0.11	0.68 ± 0.11	0.47 ± 0.16	0.55 ± 0.08
	Dist.	0.06 ± 0.01	0.16 ± 0.01	0.15 ± 0.08	0.20 ± 0.08
Lactase	Prox.	21.80 ± 3.27 ^A^	29.60 ± 3.2 ^B^	32.90 ± 3.99 ^B^	30.00 ± 6.83 ^B^
	Mid.	18.6 ± 2.99	19.20 ± 2.10	17.20 ± 5.82	20.00 ± 4.35
	Dist.	10.40 ± 3.43	8.15 ± 2.88	6.60 ± 1.05	9.62 ± 3.87
Maltase	Prox.	6.23 ± 1.49	6.23 ± 1.51	6.09 ± 3.99	6.76 ± 3.63
	Mid.	4.05 ± 1.44	5.44 ± 1.62	5.10 ± 2.69	5.27 ± 1.59
	Dist.	2.35 ± 0.48	3.38 ± 0.69	3.21 ± 0.95	3.59 ± 1.46
Dipeptidylpeptidase IV	Prox.	1.53 ± 0.98	1.70 ± 0.87	1.37 ± 0.68	2.16 ± 1.05
	Mid.	1.54 ± 1.14	2.22 ± 1.13	2.15 ± 1.77	2.47 ± 1.50
	Dist.	3.74 ± 2.20	2.80 ± 1.06	2.99 ± 1.18	2.87 ± 1.04
Aminopeptidase N	Prox.	5.81 ± 2.98	5.98 ± 2.28	8.94 ± 2.39	6.07 ± 2.46
	Mid.	5.27 ± 1.35	5.10 ± 1.32	5.16 ± 1.97	5.15 ± 1.18
	Dist.	5.69 ± 2.99	6.12 ± 2.23	6.23 ± 2.30	6.20 ± 2.28
Aminopeptidase A	Prox.	6.88 ± 2.37	7.58 ± 2.87	7.51 ± 3.11	7.64 ± 2.41
	Mid.	5.45 ± 1.46	6.88 ± 1.98	5.79 ± 2.44	7.51 ± 2.19
	Dist.	3.83 ± 0.76	4.21 ± 1.01	4.06 ± 1.18	4.40 ± 1.30

Group NC: in this control group, sows did not receive pancreatic enzymes during gestation. Group PC: in this control group, the piglets received pancreatic enzymes between 14 and 21 days of age. Group D1: in this experimental group, sows received pancreatic enzymes between 80 days of gestation and farrowing. Group D2: in this experimental group, sows received pancreatic enzymes between 30 days of gestation and farrowing. ^A,B^ Mean values marked with different letters are statistically significantly different at *p* ≤ 0.01.

**Table 14 animals-13-03448-t014:** Contents of haptoglobin and immunoglobulin IgA in the blood plasma of piglets (mg/mL).

Indices	Nutritional Group			
	NC	PC	D1	D2
Immunoglobulin IgA	0.08 ± 0.01	0.09 ± 0.02	0.90 ± 0.01	0.90 ± 0.02
Haptoglobin	0.42 ± 0.20 ^B^	0.18 ± 0.15 ^A^	0.38 ± 0.20 ^B^	0.20 ± 0.13 ^A^

Group NC: in this control group, sows did not receive pancreatic enzymes during gestation. Group PC: in this control group, the piglets received pancreatic enzymes between 14 and 21 days of age. Group D1: in this experimental group, sows received pancreatic enzymes between 80 days of gestation and farrowing. Group D2: in this experimental group, sows received pancreatic enzymes between 30 days of gestation and farrowing. ^A,B^ Mean values marked with different letters are statistically significantly different at *p* ≤ 0.01.

## Data Availability

None of the data were deposited in an official repository. Data that support the findings of this study are available upon request from the corresponding author.

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
