# Peer review of "Determining the Effect of Pancreatic-like Enzymes (PLEMs) Added to the Feed of Pregnant Sows on Fetal Size of Piglets to Minimize IUGR Syndrome Caused by Fetal Malnutrition"

_animals, 2023, doi:10.3390/ani13223448_

Round 1
Reviewer 1 Report (Previous Reviewer 1)
Comments and Suggestions for Authors
The figure legends are incomplete.
Comments on the Quality of English LanguageIt can be improved.
Author Response
Dear Reviewers,
I am enclosing answers for each of the 1, 2 and 3 Reviewers in the form of pdf files.

Reviewer 2 Report (Previous Reviewer 2)
Comments and Suggestions for Authors
Line 64-67: The original article indicates a percentage of 6 to 10%, and not 6 to 8%.
Line 67-68:The data does not have a bibliographic citation.
Line 164: The farrowing number of the sows should be mentioned, since this is directly related to neonatal mortality. That is, the sows should have been grouped by parity number for analysis.
Lines 310-359:In the explanation of results it is not mentioned which ones correspond to tables 4 and 5.
The results do not mention whether they are presented as means or standard errors.
Table 4: Data on the average weight and millimeters of back fat of sows upon entering the farrowing area should be included, as these factors can affect sow performance during lactation.
Table 5: The number of stillborn piglets intrapartum and antepartum, as well as the number of mummies, must be included, since they are important factors that show what happened during gestation and during the birthing process.
Lines 592-593 y 657-660: Considering that the enzymes that were administered are proteolytic, amylolytic and lipolytic, a possible physiological explanation must be proposed to explain how these enzymes caused the intestinal changes in the piglets. Likewise, an attempt should be made to explain how these enzymes caused greater permeability in the placenta.
It is important to ask if necropsies were performed on the piglets that died during the lactation period and, if so, why they were not related to the consumption of the enzymes.
Author Response
Dear Reviewers,
I am enclosing answers for each of the 1, 2 and 3 Reviewers in the form of pdf files.

Reviewer 3 Report (Previous Reviewer 4)
Comments and Suggestions for Authors
The authors have sufficiently addressed the comments. Therefore, I recommend the acceptance of their paper for publication in Animals.
Comments on the Quality of English LanguageA final editing regarding English language would be of value for the improvement of the text.
Author Response
Dear Reviewers,
I am enclosing answers for each of the 1, 2 and 3 Reviewers in the form of pdf files.

Round 2
Reviewer 2 Report (Previous Reviewer 2)
Comments and Suggestions for Authors
In the previous version I mentioned that the farrowing number of the sows must be included, which is directly related to the number of piglets born alive and dead. However, the requested data was not included in this new version (Muns, R., Nuntapaitoon, M., & Tummaruk, P. (2016). Non-infectious causes of pre-weaning mortality in piglets. Livestock Science, 184, 46-57;
The method and equipment with which the back fat measurement of the sows was carried out is not mentioned. Likewise, this information must be included in material and methods.
Author Response
Dear Reviewer,
I send replies to comments in the attachment

This manuscript is a resubmission of an earlier submission. The following is a list of the peer review reports and author responses from that submission.
Round 1
Reviewer 1 Report
Comments and Suggestions for Authors
Summary:
Pieszka et al. discuss the importance of microbiological enzymes with pancreatic profile to add in the feed of pregnant sows to increase their survival rate. They have done an extensive study using different several doses and different combination of feed. The main objective of this manuscript is to discuss the effect of fungal pancreatic-like enzymes of microbial origin (PLEM) on pregnant sows and they show that it
improves placental function in terms of nutrient permeability to the fetus and in
fetal development and growth while reducing the probability of the IUGR syndrome.
Major comment: It looks like all the data was dumped without giving a proper thought. Paper seems to be written in a hurry to meet a deadline.
- The introduction is extra-long (2.5 pages). They are trying to make the story, but extensive introduction doesn’t help. The results are messy and not clear. They have 14 tables and three figures which are very hard to follow. Figure legends are missing. In line 455-457 they say NC has higher plasma level ghrelin than D1, but they do not NC in figure. Controls from figures are missing. If they would have shown the data in tables in form of graphs, it would have been easy to follow.
The discussion is of 4 pages without any technical detail. “Can be concluded”, “can be further concluded” are very vague terms. The first few line in discussion (lines 471-482) should be part of introduction
Comments on the Quality of English Language
There are many repetitive words. Present study, may, might, could are all vague terms.
Reviewer 2 Report
Comments and Suggestions for Authors
Lines: 61-63. “Similar to humans, the IUGR syndrome arises spontaneously in piglets affecting 6–8%” Falta la referencia.
Lines: 61-63: “Similar to humans, the IUGR syndrome arises spontaneously in piglets affecting 6–8%”. A reference is needed for this data
Lines 64-65: A reference is needed.
Line 63-64: What is the economic cost of this problem in pigs?
Line 67: It says leavd, it should say leave
Line 177: how were the piglets euthanized?
Lines 483-493: Not a single reference is cited to support what is mentioned.
Lines: 522-556: There is too much literature cited but it is not used to contrast the results of the study.
Why were the placentas not weighed?
Why were the causes of death of the piglets not determined?
Stillborn piglets and mummies per group should have been included.
How did you verify that the administered enzymes cross the fetoplacental barrier before starting the study?
In the introduction, too much emphasis is placed on the pathology in species other than pigs. However, since the objective of the article is to solve a problem of swine production, many references that cite other species should be replaced by others specifically related to swine.
Reviewer 3 Report
Comments and Suggestions for Authors
The authors examined the effect of the prenatal administration of a mixture of pancreatic-like fungal enzymes, such as lipase, amylase, and protease, in pregnant sows at different pregnancy periods on piglets' postnatal development. The subject of this study is suitable for the “Agriculture” journal. The authors indicated that the administration of the enzyme supplement resulted in a significant shortening of gestation. Also, the pancreatic enzymes administered to sows have more live-born piglets and weaned piglets. Moreover, the authors suggest that piglets born to pancreatic-like fungal enzymes administrated to sows had higher growth rates, and they have more efficient intestinal tissue. This is an exciting study, and the authors provided new information on feeding strategies for pregnant sows regarding pig farming. I have several concerns (below indicated) about the manuscript in its current form that need to be addressed by the authors before consideration for publication. After the authors address these concerns, the manuscript can be accepted.
- The study has been well presented, but the authors should emphasize in the introduction why they chose the gestational days on which the administration was done.
-The number of animals used in the study is low. In the statistical analysis section, a power analysis showing that the number of animals is sufficient should be added. Otherwise, the reliability of the results obtained may be questioned.
-There is no analysis regarding histological imaging in the material method section. It should be clear how the authors did this.
Also;
-The results are convincing and supported by the discussion.
-The conclusions are consistent with the evidence and arguments.
-The review topic and references are appropriate.
-Tables and figures are presented clearly and understandably.
Reviewer 4 Report
Comments and Suggestions for Authors
The aim of the article ‘’ Determining the effect of pancreatic-like enzymes (PLEM) added to the feed of pregnant sows on fetal size of piglets to minimize IUGR syndrome caused by fetal malnutrition’’ was to valuate the effect of enzymes of microbiological 14 origin with pancreatic profile added to the feed of pregnant sows on foetal size in order to minimise 15 IUGR caused by foetal malnutrition. This study deals with an interesting topic and fits with the journal's scope. This is a well-designed and well-written work, providing interesting results and novel results. Therefore, I suggest the acceptance under a major revision based on the following major and minor comments.
Comments for the authors
Major comments
Abstract
- Consider providing a conclusion you have drawn from the results of your study.
Materials and Methods
- provide more data on whether the study sows received any preventive treatment.
- why were 24 sows selected to participate in the study? Has there been any sample size calculation?
Discussion:
- well structured discussion, adequately discussed the results of your study
Minor comments
- Please check the uploaded pdf file.

Comments on the Quality of English LanguageModerate editing of English language required